# In Silico and In Vitro Assessment of Carbonyl Reductase 1 Inhibition Using ASP9521—A Potent Aldo-Keto Reductase 1C3 Inhibitor with the Potential to Support Anticancer Therapy Using Anthracycline Antibiotics

**DOI:** 10.3390/molecules28093767

**Published:** 2023-04-27

**Authors:** Marek Jamrozik, Kamil Piska, Adam Bucki, Paulina Koczurkiewicz-Adamczyk, Michał Sapa, Benedykt Władyka, Elżbieta Pękala, Marcin Kołaczkowski

**Affiliations:** 1Department of Medicinal Chemistry, Faculty of Pharmacy, Jagiellonian University Medical College, Medyczna 9 St, 31-008 Cracow, Poland; marek93.jamrozik@uj.edu.pl (M.J.); michal.piotr.sapa@student.uj.edu.pl (M.S.); marcin.kolaczkowski@uj.edu.pl (M.K.); 2Department of Pharmaceutical Biochemistry, Faculty of Pharmacy, Jagiellonian University Medical College, Medyczna 9 St, 31-008 Cracow, Poland; kamil.piska@uj.edu.pl (K.P.); paulina.koczurkiewicz@uj.edu.pl (P.K.-A.); elzbieta.pekala@uj.edu.pl (E.P.); 3Department of Analytical Biochemistry, Faculty of Biochemistry, Biophysics and Biotechnology, Jagiellonian University, Gronostajowa 7 St, 31-007 Cracow, Poland; benedykt.wladyka@uj.edu.pl

**Keywords:** anthracyclines, enzymatic inhibitors, drug resistance, cardioprotection, docking, molecular dynamics, ASP9521

## Abstract

Anthracycline antibiotics (ANT) are among the most widely used anticancer drugs. Unfortunately, their use is limited due to the development of drug resistance and cardiotoxicity. ANT metabolism, performed mainly by two enzymes—aldo-keto reductase 1C3 (AKR1C3) and carbonyl reductase 1 (CBR1)—is one of the proposed mechanisms generated by the described effects. In this study, we evaluated the CBR1 inhibitory properties of ASP9521, a compound already known as potent AKR1C3 inhibitor. First, we assessed the possibility of ASP9521 binding to the CBR1 catalytic site using molecular docking and molecular dynamics. The research revealed a potential binding mode of ASP9521. Moderate inhibitory activity against CBR1 was observed in studies with recombinant enzymes. Finally, we examined whether ASP9521 can improve the cytotoxic activity of daunorubicin against human lung carcinoma cell line A549 and assessed the cardioprotective properties of ASP9521 in a rat cardiomyocytes model (H9c2) against doxorubicin- and daunorubicin-induced toxicity. The addition of ASP9521 ameliorated the cytotoxic activity of daunorubicin and protected rat cardiomyocytes from the cytotoxic effect of both applied drugs. Considering the favorable bioavailability and safety profile of ASP9521, the obtained results encourage further research. Inhibition of both AKR1C3 and CBR1 may be a promising method of overcoming ANT resistance and cardiotoxicity.

## 1. Introduction

Anthracycline antibiotics (ANT) are a group of therapeutics widely used in the treatment of various types of cancers, both solid tumours and haematological malignancies [1]. The history of this class of drugs dates to the 1960s, when the first compounds were isolated from the *Streptomyces* genus. Since then, several natural and synthetic ANT have been introduced to therapy as effective cell-cycle nonspecific compounds, which act mainly via the inhibition of topoisomerase II and intercalation into double-strand DNA [2]. Doxorubicin (DOX) and daunorubicin (DNR) are some of the most widely used ANT. Unfortunately, their usage is limited due to drug resistance developing during the therapy and due to cardiotoxicity, which can manifest even several years after the treatment period [3]. Researchers and physicians are continually trying to extend the knowledge about the reasons behind the side effects of ANT to apply proper therapeutical solutions. One of the approaches is adjuvant therapy with either already registered drugs or new molecules, leading to improved ANT efficacy and safety [3,4]. To date, the only approved drug targeting the safety issues of ANT is dexrazoxane. The compound acts by preventing ANT–iron binding and the subsequent formation of reactive oxygen species [5]. However, the exact molecular mechanisms of ANT side effects remain unclear.

One of the main new hypotheses explaining the limited use of ANT points out a crucial role of the activity of ANT metabolic enzymes, leading to the formation of secondary alcohols by the reduction of carbonyl moiety in the ANT C13 position [6]. In a series of in vitro experiments, metabolites of two main ANT–doxorubicinol (DOXol) and daunorubicinol (DNRol)–possessed a significantly weaker cytotoxic effect with increased cardiotoxic effect [7].

There are two main groups of enzymes leading to the formation of C13-reduced ANT metabolites: aldo-keto reductases (AKR) and carbonyl reductases (CBR). Enzymes belonging to both groups are monomeric, cytosolic, NAD(P)H-dependent proteins responsible for the two-electron reduction of various endogenous substrates, including prostaglandins, quinones and steroids [8,9]. Among these enzymes, carbonyl reductase 1 (CBR1) and aldo-keto reductase 1C3 (AKR1C3) are considered the most important for ANT C13 reduction [10]. CBR1 is an enzyme responsible for the metabolism of several xenobiotics, including warfarin, nabumetone, and bupropion, whereas the most pharmacologically important AKR1C3 action is the conversion of dehydroepiandrosterone and androstenedione into 5-androstenediol and testosterone, respectively [9,11]. Several scientific articles suggest that inhibition of either CBR1 or AKR1C3 can be a key component in overcoming ANT resistance and cardiotoxicity [12,13,14,15]. A potent CBR1 inhibitor—hydroxy-PP-Me—enhanced the chemotherapeutic effect of DOX in breast cancer cells and prevented cardiotoxicity in MDA-MB-157-implanted tumour mice [12]. Amide alkaloid piperlongumine enhanced the cytotoxic effect of DOX in DU-145 prostate cancer cells [16]. Similar experiments have been performed with inhibitors of a second major ANT reductase, AKR1C3. Drugs such as dinaciclib, olaparib, and roscovitine reversed ANT resistance mediated by AKR1C3 activity and increased DNR cytotoxicity in several cellular models [14,17,18]. A group of selective AKR1C3 inhibitors, built on a cinnamic acid scaffold, has been tested together with DNR on multiple acute myeloid leukaemia cells, resulting in significant enhancement of the anticancer activity of the drug [15]. Although the search for either CBR1 or AKR1C3 inhibitors is being increasingly popular in the context of overcoming unfavourable properties of ANT, compounds targeting both the main reductases responsible for ANT metabolism have not yet been described. Thus, it is unclear whether this dual inhibitory effect would result in an even better improvement of ANT activity and safety than in the case of selective enzymatic inhibition.

ASP9521 is a potent indole-based AKR1C3 inhibitor. Considering nanomolar activity of ASP9521 against AKR1C3 and thus the possible inhibition of testosterone synthesis, the compound was tested in the treatment of castrate-resistant prostate cancer. It successfully passed in vitro and preclinical development evaluation and entered into clinical development but was then withdrawn due to a lack of clinical efficacy. However, the clinical evaluation indicated its acceptable safety and tolerability profile [11,19]. Structurally, ASP9521 possesses similar features to known CBR1 inhibitors, e.g., luteolin and cinnamamide derivatives, which have previously been described as compounds potentiating ANT activity against cancer cells (Figure 1) [13,20]. Fragments containing H-bond-accepting carbonyl groups (e.g., amide, ketone), H-bond donors (hydroxy or amine groups), and aromatic rings are considered to be crucial for CBR1 ligand recognition. The same features were found in similar positions of ANT and the CBR1 substrates. Additionally, a piperidine ring present in both ASP9521 and cinnamamide Compound 15 was assessed as a significant feature improving inhibitory activity against CBR1 within a group of cinnamamide derivatives [13]. Those findings suggest that ASP9521 may inhibit the activity of both crucial ANT reductases—AKR1C3 and CBR1—and this dual mode of action may be desirable in overcoming adverse effects caused by ANT. Additional premises were the similarities observed between the CBR1 and AKR1C3 catalytic sites. In the literature, the binding mode of CBR1 inhibitors, like cinnamamides, flavonoids, and piperlongumine, was in line with the binding mode of indole-based AKR1C3 inhibitors within their own catalytic sites. Two molecules structurally similar to ASP9521 have been crystallised with AKR1C3 in its catalytic site (PDB codes 4WDW and 4WDX) and formed H-bonds with amino acid residues responsible for the catalytic reaction (Tyr55 and His119) [21]. In the vicinity of those residues, there were plenty of amino acid side chains with aromatic properties (phenylalanines, tyrosines, tryptophans), which explains the importance of aromatic fragments in AKR1C3 ligands. The same types of interactions have been described as crucial in terms of potential CBR1 inhibitors, where H-bonds were formed with amino acids involved in catalytic reaction: Ser139 and Tyr193, and the binding pose was stabilised via a hydrophobic π-π interaction with Trp229 [20].

Taking into account the above facts, in the present study, we focused on the assessment of the potential CBR1 inhibitory activity of ASP9521. We hypothesised that ASP9521 can be an inhibitor of both main ANT reductases, making it a molecule with an as-yet undescribed mechanism of action. First, we docked ASP9521 to CBR1 to assess whether the compound can form interactions that are important for ligand recognition. We compared the binding mode with the one observed for known CBR1 ligands. Then, we performed an MD simulation to check whether ASP9521 holds the interactions observed in docking through a 20 ns simulation period and to evaluate the stability of the ligand within the CBR1 catalytic site. After in silico evaluation, the inhibitory activity was assessed in studies with recombinant CBR1 proteins, and then, the cytotoxic effect of simultaneous administration of DNR and ASP9521 was determined on A549 cancer cells. Finally, ASP9521 was examined as a potential cardioprotective agent preventing rat cardiomyocyte H9c2 cells from damage caused by DNR or DOX.

## 2. Results and Discussion

### 2.1. Molecular Docking

To evaluate whether ASP9521 has the potential to bind to CBR1, we initially performed molecular docking to the CBR1 model. The ligand was docked successfully and its binding mode within the CBR1 catalytic site was in line with the one described for known CBR1 ligands (Figure 2), including luteoline, piperlongumine, hydroxy-PP, and ANT–DOX and DNR [16,22]. The amide group of ASP9521 was placed in the vicinity of two out of four amino acid residues constituting the CBR1 catalytic tetrad: Ser 139 and Tyr193, side chains of which formed H-bonds with the carbonyl oxygen atom. In terms of ANT, the same residues formed H-bonds with the carbonyl group, which underwent a reduction to a hydroxy group. A flat indole ring of ASP9521 was placed in the vicinity of Trp229, forming π-π interaction with its side chain. In terms of ANT, this space was occupied by an anthraquinone-based tetracyclic fragment, which formed analogous interactions. This molecular interaction was also observed for other CBR1 ligands, including hydroxy-PP present in CBR1 crystal structures 1WMA, 3BHJ, and 3BHM [23,24]. The last interaction observed between CBR1 and ASP9521 was the H-bond formed between the ligand hydroxy group and the backbone of Met234. This amino acid residue, together with an adjacent Ala235, is supposed to play a key role in forming H-bond interactions with hydroxy and/or amine groups present in sugar fragments of various ANT. Unlike Met234, we did not observe any interactions between ASP9521 and Ala235. The piperidine ring of ASP9521, substituted with 2-hydroxy-2-methylpropyl moiety, has shown a similar position to the one from cinnamamide derivative Compound 15, described previously as a promising adjuvant of ANT therapy [13].

### 2.2. Molecular Dynamics of CBR1-ASP9521 Complex

Molecular docking revealed the potential of ASP9521 to be a CBR1 ligand, thus we decided to confirm ASP9521 binding mode durability in MD simulation. The best-scored CBR1-ASP9521 complex obtained in molecular docking (glide gscore = −7.193 kcal/mol) was used in a 20 ns MD simulation to assess ligand stability within the CBR1 catalytic site. For this purpose, we measured distances between amide oxygen and carbon atoms of ASP9521 and two elements of the CBR1 enzymatic complex responsible for proton transfer to the CBR1 substrate. The latter elements were the phenol group of the Tyr193 side chain and the C4 pyridine atom of NADPH [25]. Throughout the whole simulation, ASP9521 remained within the CBR1 catalytic site, forming stable interactions with the two abovementioned important amino acid residues, Ser139 and Tyr193. During the 20 ns of the simulation, the average distances between the aforementioned pairs of elements were 2.99 Å and 3.48 Å, respectively (Figure 3). The NADPH molecule kept its position relatively constant throughout the simulation. In terms of Tyr193, after 17 nanoseconds of the simulation, its side chain flipped towards the NADPH ribose ring, resulting in an increase in the distance between the amino acid residue and ASP9521 from about 3 Å to about 4–6 Å. Overall, throughout the simulation, ASP9521 remained stable and close to the key components of the enzyme complex.

The Simulation Interactions Diagram tool detected and recorded all protein–ligand interactions during the 20 ns of the simulation (Figure 4).

The most important interactions for a CBR1 ligand recognition–H-bonds with Ser139 and Tyr193–have been observed for most of the simulations time. The interaction between the amide oxygen atom of ASP9521 and the side chain of Ser139 was observed for 92.9% (1858/2000 frames–fr.) of the simulation time; and H-bond with Tyr193 for 87.4% (1748 fr.). These observations support the hypothesis that the interactions with ASP9521 remain stable within the CBR1 catalytic site and that the molecule does not dissociate when the protein–ligand complex undergoes dynamic changes over time. Unlike molecular docking, MD almost did not evidence the presence of interaction with a Met234 backbone. This interaction was observed only 1% (20 fr.) of the time—at the very beginning of the simulation. However, the hydroxy group of ASP9521 was more likely to interact with Met234 through a water bridge. This type of interaction was observed for 12% of the simulation time (238 fr.). This observation suggests that we cannot completely exclude the potential importance of Met234 in the binding mode of ASP9521 within CBR1. 

Instead of an H-bond with Met234, MD revealed another amino acid residue forming an interaction with the hydroxy group of 2-hydroxy-2-methylpropyl moiety of the ligand. For 86.2% of the simulation time (1724 fr.), we observed an H-bond with a backbone of Phe94. This observation suggests a second possible binding mode of ASP9521 within the CBR1 catalytic site in which the ligand does not occupy a space in the vicinity of Met234 and Ala235 (as observed in the case of ANT), but instead, a piperidine ring with 2-hydroxy-2-methylpropyl moiety adopts a conformation, which forms an interaction with the Phe94 backbone (Figure 5a).

We additionally measured and compared distances between the ASP9521 hydroxy group and the backbone oxygen atoms of Met234 and Phe94. At the beginning of the simulation, the distance to Met234 was 2.69 Å, while the average distance during the simulation was 5.29 Å. Oppositely, the initial distance to Phe94 was 4.22 Å, while the average distance during the simulation was 3.05 Å (Figure 5b), which confirms that the piperidine ring with its aliphatic moiety preferred to be positioned closer to Phe94.

The location of the ASP9521 piperidine ring with attached 2-hydroxy-2-methylpropyl moiety, forming H-bonds mainly with Phe94 instead of Met234, was the most substantial difference between the binding modes observed in docking and MD simulation. In the first nanosecond of the MD simulation, the whole fragment shifted towards the backbone of Phe94, and the conformation of ASP9521 was then similar to that observed in the AKR1C3 crystal structure 4WDX, with another indole-based derivative co-crystallised within the catalytic site of the enzyme. The H-bond formation with Phe94 backbone was intriguing, as this residue has been indicated as one of the interaction points between CBR1 and glutathione molecules in crystal structures 3BHJ and 4Z3D [23,26]. Glutathione is suggested to play an important role within the active site, facilitating the correct arrangement of small CBR1 substrates (e.g., quinones). Additionally, it is suggested that glutathione protects the catalytic centre of CBR1 through a switch-like mechanism and its conformation varies according to the presence of the ligands in the vicinity of the enzyme [26]. Our MD simulation of CBR1-ASP9521 complex revealed that this glutathione region can be an important point of interaction with enzymatic inhibitors. Considering both docking and MD, we observed two possible binding modes of ASP9521—the one more similar to those observed for ANT, flavonoids, and cinnamamides (obtained by molecular docking) and another one, in which the ligand conformation is similar to the conformation observed in the AKR1C3 4WDX crystal structure (obtained via MD simulation). In both cases, we detected target–ligand interactions with amino acid residues mentioned previously as essential for the binding of other enzyme ligands.

MD simulation also revealed that four amino acid residues have been involved in forming hydrophobic interactions with the indole ring of ASP9521. The π-π stacking with Trp229, formed in the initial CBR1-ASP9521 complex resulting from docking, was observed for 11.6% of the simulation time (232 fr.; its duration was shorter than we could expect based on the ligand structure and results of the docking). That interaction was supported by additional van der Waals contacts with hydrophobic side chains within 3.6 Å of ligand aromatic or aliphatic carbon atoms, present 77% of the time (1347 fr.). The same type of hydrophobic interaction has been detected with residues next to Ser 139: Ile140 (72.1% of the time; 1442 fr.) and Met141 (45.4% of the time; 908 fr.). Based on the MD results, we may assume that such locations of Ile140, Met141, and Trp229 and observed hydrophobic interactions stabilise the aromatic part of ASP9521 and facilitate proper positioning of the whole ligand within the CBR1 catalytic site. 

The π-cation interaction between the indole ring of ASP9521 and the side chain of Arg144 was detected for 17% of the simulation time (343 fr.). Additionally, Arg144 was involved in the formation of a water bridge with an oxygen atom of the methoxy group attached to the indole ring of ASP9521 (27.2% of the time; 539 fr.).

### 2.3. Assessment of Human Recombinant CBR1 Inhibition by ASP9521

We verified the hypothesis and observations obtained through in silico simulations using an in vitro CBR1 inhibitory assay measuring the change in the rate of CBR1 enzymatic activity upon the addition of ASP9521. The determined ASP9521 IC_50_ value of 44.00 μM confirmed its moderate inhibitory properties against the enzyme. The obtained value is higher than those for the most potent CBR1 inhibitors (IC_50_ of flavonoids and 8-hydroxy-2-iminochromene derivatives around 0.1–0.4 μM), nevertheless, the report on the dual AKR1C3 and CBR1 inhibitory activity of ASP9521 sheds new light on its mechanism of action as a representative of multi-functional compounds as well as on the issue of ANT metabolism and directly related risks [20,27]. ASP9521 was designed to be a potent AKR1C3 inhibitor, and to the best of our knowledge, there are no data about its activity towards other molecular targets. Thus, we believe that further optimization of its structure would result in structures with more balanced AKR1C3-CBR1-inhibitory properties. It remains unclear whether ASP9521 conformation observed during the MD simulation and its shift towards Phe94 instead of Met234 is either more preferable or undesirable in terms of ligand affinity for CBR1. Studies on flavonoid compounds revealed that the ability to interact with Met234 improved significantly the inhibitory potency (IC_50_ for flavone: over 10 μM vs. IC_50_ for 7-hydroxyflavone: 0.6 μM; in the case of the second compound, the 7-hydroxy group formed a H-bond with Met234), but we do not have such analysis for compounds targeting interaction with Phe94 [20]. This information would be essential for a rational design of new CBR1 inhibitors and dual CBR1-AKR1C3 inhibitors.

### 2.4. Assessment of ASP9521 Co-Administration Together with DOX/DNR on ANT Anticancer and Cardiotoxic Properties

To verify the hypothesis that co-administration of ASP9521 can be beneficial for ANT treatment, we performed experiments on a human lung carcinoma cell line A549 and rat cardiomyocyte H9c2. Cell line selection was based on literature research confirming high expression levels of CBR1 and AKR1C3. Increased levels of either one or both of the mentioned enzymes have been shown to correlate with greater resistance of cells to DOX and DNR as well as with the increased risk of cardiotoxicity [28]. 

We first examined whether ASP9521 (25 μM) enhances the anticancer activity of DNR (0.05–1 μM) in an A549 cells model. ASP9521 used alone did not possess any cytotoxic activity. However, co-administration of ASP9521 with DNR, statistically improved the anticancer activity of the drug after 48 h of incubation (Figure 6). This enhancement was observed for most DNR concentrations used in the experiment, except 0.05 μM. Additionally, we calculated and compared IC_50_ of DNR used separately or in combination with ASP9521 (25 μM). Obtained IC_50_ values were 0.442 μM (separately) and 0.379 μM (in combination with ASP9521). Therefore, ASP9521 led to a decrease in IC_50_ of 14.25%. These results confirm that the inhibition of ANT metabolism is a significant factor for effective anticancer properties. Previously, a number of studies have shown a relationship between increased expression of CBR1 and AKR1C3 in cancer cells and reduced anticancer activity of ANT. It is estimated that the concentration of DOXol, causing the death of 50% of cancer cells, is 50–150 times higher than for DOX [28]. Similar results were observed for DNR and its metabolite. Studies of cells taken from leukaemia patients have shown a relationship between increased CBR1 expression and reduced response to DNR treatment [7].

The second part of in vitro assessment was the evaluation of the cardioprotective properties of ASP9521 (25 μM; Figure 7). H9c2 cells were preincubated with the tested compound for 3 h and then treated with 1 μM DOX or DNR for next 24 h. Compared to the control (cells treated neither with ASP9521 nor ANT), the viability of the cells treated with DNR was 60%. Additional pre-incubation with ASP9521 resulted in increased cell viability to 73%. The results of experiments with DOX were in line with the ones observed when DNR was applied. The viability of H9c2 cells treated with DOX was 79%, and that of those which were additionally preincubated with ASP9521 was 93%. This means that additional administration of the tested compound almost entirely protected cardiomyocytes from the toxic activity of DOX. These observations correspond with the results of other studies on H9c2 cells, including the evaluation of the cardioprotective effect of cinnamamide derivatives [29]. The reasons behind the evaluation of CBR1/AKR1C3 inhibitor as a cardioprotective agent came from considerable research, which confirmed the harmful effect of ANT-reduced metabolites on cardiac tissue. In experiments carried out on isolated rabbit hearts, it was shown that DOXol reduced the contractile capacity of the heart 30 times more than DOX. Apart from their impact on calcium homeostasis, reduced ANT metabolites are believed to inhibit the activity of pumps regulating the intracellular concentrations of sodium, potassium, and magnesium ions [30,31]. Experiments in mice showed that overexpression of CBR1 accelerated the development of DOX-induced cardiotoxicity. In contrast, mice lacking the CBR1 gene did not show signs of cardiotoxicity [32,33].

Based on the results of the enzymatic CBR1 inhibitory assay, we may assume that AKR1C3 inhibition was mainly responsible for the observed improvement in DNR activity as well as the cardioprotective effect and that it outweighs the effect associated with CBR1 inhibition. However, future design and development of dual CBR1-AKR1C3 inhibitors should lead to a more precise assessment of dual inhibitory strategy in the context of potentiation of ANT activity.

## 3. Materials and Methods

### 3.1. Software Used for In Silico Simulations and Reagents Used in In Vitro Experiments

Small-molecule Drug Discovery Suite with Desmond GPU (Schrödinger, Inc., New York, NY, USA) was licensed for Jagiellonian University Medical College. ASP9521 was purchased from Tocris (Bio-Techne brand; Bristol, UK). NADPH and DNR were purchased from BIOKOM. The sulforhodamine B cytotoxicity assay kit was purchased from Merck (Darmstadt, Germany).

### 3.2. Molecular Docking

Molecular docking was performed using Small-Molecule Drug Discovery Suite (Schrödinger, Inc., New York, NY, USA). The structure of ASP9521 was prepared using LigPrep tool (Schrödinger Release 2018–3: LigPrep, Schrödinger, Inc., New York, NY, USA). An energy-minimized 3D conformer in a proper protonation state predicted in the pH range 7.4 ± 0.2 was generated. The CBR1 model used for docking studies (based on the PDB structure encoded 1WMA) was prepared as described previously and characterized by the following enrichment parameters: AUC = 0.87, BEDROC_α=20_ = 0.552 [13]. The model consisted of CBR1 protein and enzymatic cofactor dihydro nicotinamide adenine dinucleotide phosphate (NADPH). The molecular docking was performed using Glide SP mode, with an H-bond constraint set on Ser139 [34]. Docking was performed with a rigid enzyme and flexible ligand. Post-docking minimization retrieved 5 poses, which were evaluated based on glide gscore function value and visual inspection of the obtained target–ligand interactions (mainly H-bonds and π-aromatic interactions).

### 3.3. Molecular Dynamics and Subsequent Analysis of CBR1-ASP9521 Complex

The best-scored CBR1-ASP9521 complex obtained via molecular docking (glide gscore = −7.193 kcal/mol) was chosen to undergo MD. The simulation was performed using Desmond GPU (Desmond Molecular Dynamics System, D. E. Shaw Research, New York, NY, USA, 2018. Maestro-Desmond Interoperability Tools, Schrödinger, New York, NY, USA, 2018). A system for the simulation was prepared using the System Builder module. The CBR1-ASP9521 complex was placed in an orthorhombic box, the size of which was calculated using the buffer method, with the box boundaries positioned 10 Å from the complex in all dimensions. TIP4P solvent model was applied, and 0.15 M of sodium and chlorine ions was added to simulate a cellular environment within the box. After initial model relaxation, performed in a default mode, a 20 ns simulation was run utilizing the OPLS3e force field and NPT ensemble. The trajectory interval was set to 10 ps, meaning that every 10 ps, an intermediate state (frame) of CBR1-AKR1C3 complex was captured and saved. As an output, 2000 CBR1-ASP9521 frames were obtained and analysed using the Simulation Event Analysis tool to measure distances between key CBR1 elements—phenol group of Tyr193 side chain and C4 pyridine atom of NADPH—and selected ASP9521 atoms—amide oxygen atoms and amide carbon atoms, respectively. Those pairs of characteristics are crucial considering the CBR1 mechanism of action. Simulation Event Analysis was also used for measuring distances between the ASP9521 hydroxy group from 2-hydroxy-2-methylpropyl moiety and backbone oxygen atoms of two amino acid residues—Met234 and Phe94—to assess which residue is more likely to contribute to interactions with the ligand. The Simulation Interactions Diagram tool was used to assess the types and durations of molecular interactions between ASP9521 and CBR1 amino acid residues formed within the 20 ns simulation period: ionic bonds, H-bonds and water bridges, and π-aromatic interactions.

### 3.4. Determination of CBR1 Inhibitory Properties Using Human Recombinant Enzyme

Expression and purification of human recombinant CBR1 were performed in *Escherichia coli* Rosetta (DE3) and described in detail previously by Koczurkiewicz-Adamczyk et al. [29]. The initial CBR1 inhibition tests were performed at 100 μM of ASP9521 After observing the inhibitory activity in that concentration, to determine IC_50_ value, additional assessments were performed in the concentration range 0.1–100 μM (0.1, 1, 10, 25, 50, 100 μM). Incubation mixtures containing recombinant CBR1 (final concentration 0.5 µM), menadione (120 μM) and ASP9521 or vehicle in phosphate buffer (pH 7.4) were preincubated for 5 min in 37 °C on UV-transparent 96-well microplate. Next, NADPH solution was added to a final concentration of 200 μM to initiate the reaction. The reaction was monitored using absorbance measurement at λ = 340 nm for 10 min in a microplate reader (SpectraMax iD3, Molecular Devices, San Jose, CA, USA). A decrease in absorbance, corresponding to NADPH oxidation, was used to determine the initial velocities of reactions via a linear regression method. The inhibitory properties of ASP9521 were determined by comparing the velocity of the reaction against vehicle control (the percentage inhibition was calculated considering the activity in the absence of inhibitors to be 100%). DMSO concentration <1%, did not affect enzyme activity. Each experiment was performed in three repetitions.

### 3.5. Cell Cultures

Human lung carcinoma cell line A549 (ATCC CCL-18) and rat cardiomyocyte H9c2 (ATCC CRL-1446) were cultured in appropriate culture media recommended by ATCC and supplemented with 10% foetal bovine serum (FBS; Gibco, Life Technologies, Carlsbad, CA, USA) and 1% antibiotic mixture (Gibco, Life Technologies, Carlsbad, CA, USA). The cells were cultured in a standard condition of temperature (37 °C) and CO_2_ concentration (5%).

ASP9521 was applied from DMSO stock solutions and diluted in the culture media to the working concentrations. DOX/DNR were applied to the culture media diluted to the working concentrations from a freshly made stock solution in DMSO (100 μM).

### 3.6. Cytotoxicity Analysis

For evaluation of the chemosensitising effect of ASP9521, sulforhodamine B (SRB) assay was used. Cells were seeded in 96well plates at a density of 1 × 10^4^ per well. After 24 h, cells were incubated with DNR in the concentration range of 0.05–1 μM (0.05; 0.1; 0.25; 0.5; 1 μM) and ASP9521 (25 μM) or a vehicle. After the next 48 h, an SRB assay was performed. Cells were fixed with 50% trichloroacetic acid at 4 °C for 1 h. Next, they were washed with water and stained with SRB solution. After 20 min, cells were washed 4 times with 1% acetic acid, and the remaining SRB was dissolved in a solubilization solution (10 mM Tris base solution). Absorbance was read at 565 nm (SpectraMax iD3, Molecular Devices, San Jose, CA, USA), and viability percentage was calculated by dividing the absorbance of experimental wells by the absorbance of the control (×100%). Three separate experiments were performed.

For evaluation of the cardioprotective effect of ASP952, 3-(4,5-dimethylthiazol-2-yl)-2,5-diphenyltetrazolium bromide (MTT) assay was used. Cells were seeded in 96-well plates at a density of 1 × 10^4^ per well. After 24 h, cells were preincubated for 3 h with ASP9521 (25 μM). Then, the culture medium was changed, and cells were incubated with DOX/DNR (1 μM). Control cells were incubated in culture medium without analysed compounds. An MTT assay was performed as described previously [35]. The viability of the cells incubated with DOX/DNR or additionally pre-incubated with ASP9521 was presented as a percentage of viability compared to the control, considering the viability in the absence of compounds to be 100%.

The selection of concentrations of ASP9521, DOX, and DNR was based on the experimental data published previously [13,29].

### 3.7. Statistical Analysis

The independent Student *t*-test was used to check the statistical significance of the difference between the IC_50_ values for DNR and the combination of DNR with ASP9521 (A549 cells only). One-way ANOVA with Tukey’s post-hoc test was used to check the statistical significance of the difference in viability cells as a result of adding 25 μM ASP9521. The calculations were carried out in the GraphPad Prism program. Statistical significance was determined for the significance level of *p* < 0.05.

## 4. Conclusions

The observed structural features of ASP9521 predisposed the compound to be considered as a potential CBR1 inhibitor. Molecular docking and MD simulation revealed two possible binding modes of the compound within the CBR1 catalytic site, both implying its highly probable affinity. Further research is needed to assess whether interaction with either Met234 or Phe94 is preferable in terms of ligand recognition and ability to effectively inhibit the activity of the CBR1 enzyme. Moderate inhibitory activity of ASP9521 was confirmed in biological studies with recombinant CBR1. The obtained results, together with the previous evaluation of ASP9521, allowed us to define it as a dual AKR1C3-CBR1 inhibitor—a compound that inhibits both main reductases responsible for ANT metabolism. Such a cumulative and not-yet-described mechanism of action can be promising in terms of overcoming ANT drug resistance and reducing the cardiotoxicity of this group of drugs. However, further development of compounds with more balanced inhibitory properties between both AKR1C3 and CBR1 is necessary to confirm this hypothesis. ASP9521 improved the cytotoxic activity of DNR on human lung carcinoma cell line A549. The compound also prevented rat cardiomyocytes H9c2 from the toxic activity of DOX, DNR or their reduced metabolites. Taking into account that the compound was previously defined to be safe in the clinical trial and also bearing in mind its good bioavailability (after oral administration), ASP9521 remains a promising molecule for evaluation not only in monotherapy of cancer but also as adjuvant therapy with ANT.

## Figures and Tables

**Figure 1 molecules-28-03767-f001:**
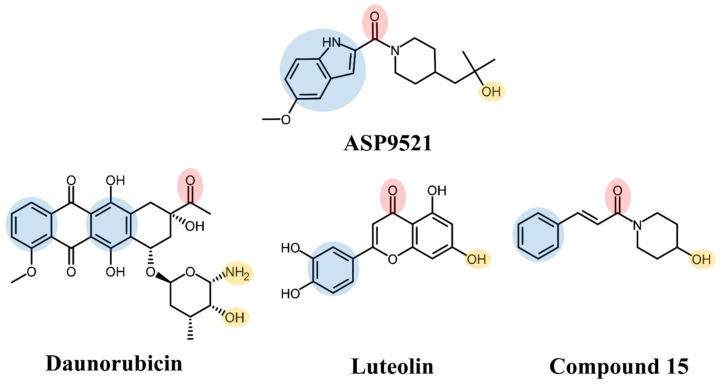
Structures of ASP9521 and selected known CBR1 ligands: daunorubicin (substrate), luteolin and cinnamamide-derivative Compound 15 (both inhibitors); the circles indicate structural similarities that may be the features important for CBR1 binding: H-bond acceptor (red), H-bond donor (yellow), hydrophobic aromatic ring (blue).

**Figure 2 molecules-28-03767-f002:**
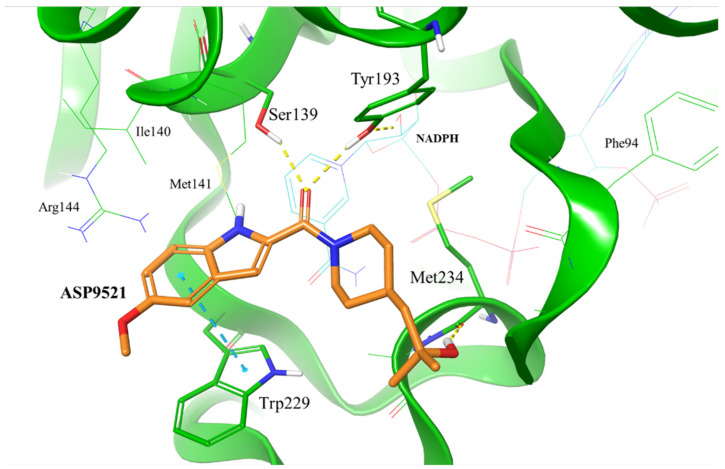
Putative binding mode of ASP9521 (orange) within CBR1 catalytic site obtained by molecular docking. Amino acid residues engaged in ligand binding throughout H-bonds (dotted yellow lines) and π-π stacking (dotted blue lines) displayed as thick sticks; NADPH and amino acid residues not involved directly in ligand binding but were pointed out as important for ligand binding in a subsequent molecular dynamics simulation (MD), displayed as thin sticks.

**Figure 3 molecules-28-03767-f003:**
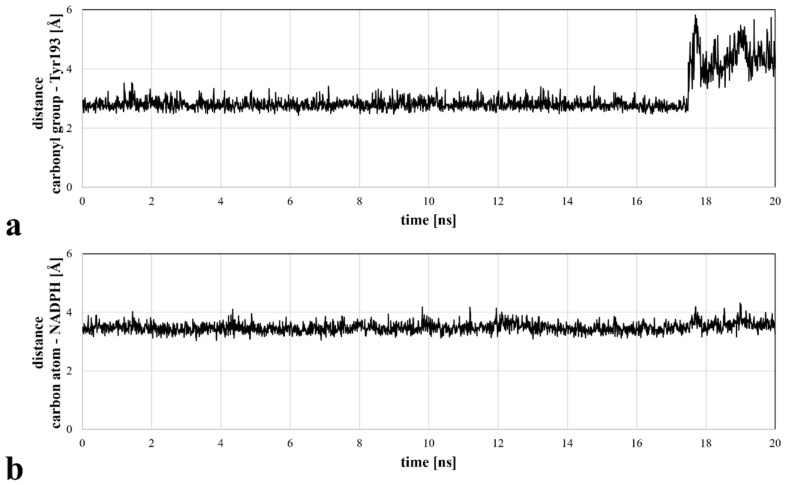
Distances between the selected ASP9521 atoms and substantial CBR1 enzymatic complex elements, measured during a 20 ns MD simulation: (**a**) distance between the amide oxygen atom of ASP9521 and Tyr193 side chain phenol oxygen atom of CBR1; (**b**) distance between an amide carbon atom of ASP9521 and C4 pyridine atom of CBR1.

**Figure 4 molecules-28-03767-f004:**
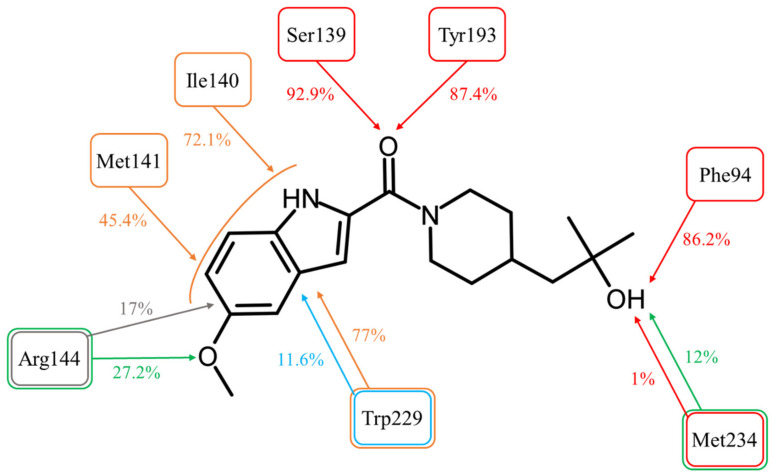
Molecular interactions between ASP9521 and CBR1 amino acid residues observed during 20 ns of MD simulation (expressed as a percentage of the entire simulation time in which they have been observed). The colours of frames and arrows represent types of interactions: red—H-bond; green—water bridge; blue—π-π stacking; grey—π-cation; orange—other hydrophobic interactions.

**Figure 5 molecules-28-03767-f005:**
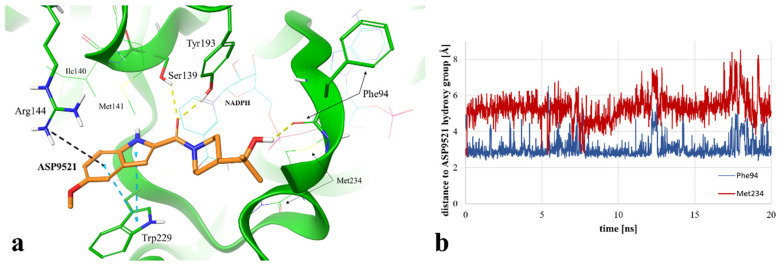
(**a**) Exemplary binding mode of ASP9521 (orange) within the CBR1 catalytic site obtained via MD simulation (captured at 17.23 ns from the start of the simulation). Amino acid residues engaged in ligand binding throughout H-bonds (dotted yellow lines), π-π stacking (dotted blue lines), and pi-cation interaction (dotted grey lines) are displayed as thick sticks; NADPH and amino acid residues not involved directly in ligand binding in the selected timepoint but determined during the simulation as important for ligand binding to the active site are displayed as thin sticks; water molecules and water bridges are not displayed; (**b**) distances between the hydroxy group of ASP9521 and backbone oxygen atoms of Phe94 (blue line) and Met234 (red line), measured during a 20 ns MD simulation.

**Figure 6 molecules-28-03767-f006:**
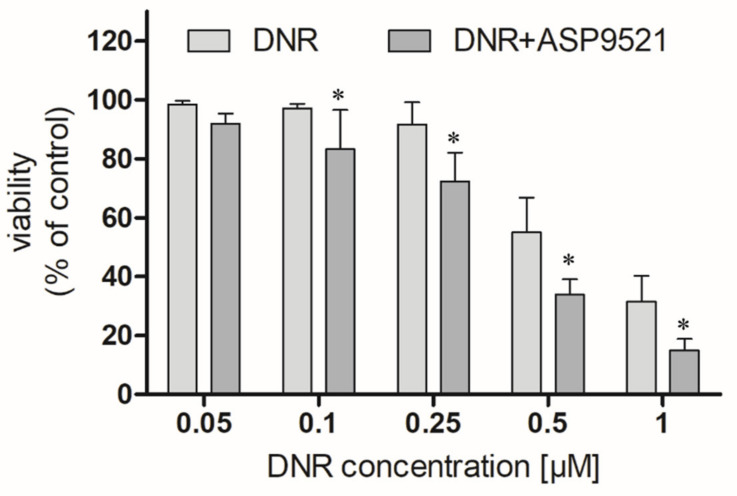
Effect of the combined administration of ASP9521 (25 μM) and DNR on the viability of A549 cells after 48 h of incubation compared to DNR without the addition of the ANT reductase inhibitor; * statistical significance −/+ ASP9521 at *p* < 0.05.

**Figure 7 molecules-28-03767-f007:**
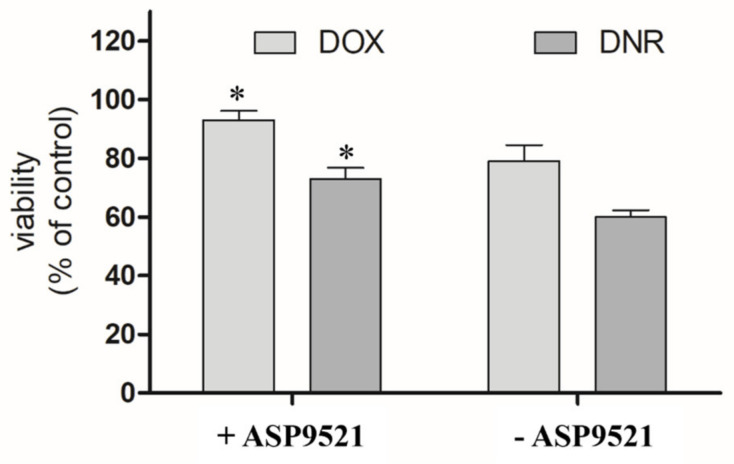
Effect of ASP9521 (25 μM) on DOX/DNR-induced damage prevention in H9c2 cells. H9c2 cells were preincubated in the presence of ASP9521 (25 μM) for 3 h. Then, DOX/DNR were added for the next 24 h. Viability of cells was determined using an MTT assay. The graph represents average cell viability expressed as percentage of control (no treated cells) ± SEM of three replications of three independent experiments; * statistical significance −/+ ASP9521 at *p* < 0.05.

## Data Availability

The initial CBR1 structure was downloaded from the PDB database (code 1WMA–https://www.rcsb.org/structure/1WMA lately accessed on 22 March 2023). The optimized CBR1 model is available on request from the corresponding author.

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
