# Peer review of "In Silico and In Vitro Assessment of Carbonyl Reductase 1 Inhibition Using ASP9521—A Potent Aldo-Keto Reductase 1C3 Inhibitor with the Potential to Support Anticancer Therapy Using Anthracycline Antibiotics"

_molecules, 2023, doi:10.3390/molecules28093767_

Round 1
Reviewer 1 Report
See file attached

Reviewer 2 Report
In this manuscript by Jamrozik et al., the authors carry out an assessment of carbonyl reductase 1 inhibition by ASP9521, a compound formerly known as potent AKR1C3 inhibitor, by means of molecular modeling and in vitro assays. Overall, the manuscript is well-written, well-organized, and easy to understand. The results are presented properly and the discussions are given thoroughly. I appreciate this work and recommend it for publication in Molecules.
Reviewer 3 Report
A very nice and thorough study, if somewhat limited in scope, since only one compound is investigated.
I recommend publication as is, but have found one mistake on line 308:
It should be 14.25%, not 14,25%.
Reviewer 4 Report
In this manuscript, the authors employed molecular docking to generate the docking pose of ASP9521 in CBR1 catalytic site and used molecular dynamic simulation to evaluate the stability of this pose. Then, the authors measured the enzymatic inhibition activity against CBR1. The study showed that ASP9521 improved the anti-cancer activity of daunorubicin. This study offered a possibility for drug combinations to overcome ANT resistance and cardiotoxicity.
Here, I have several questions and suggestions for the authors:
1, Did molecular docking generate other binding poses of ASP9521 in CBR1? They are not the top 1 pose based on the docking score. But they might offer more poses submitted to MD simulation.
2, Could the authors superimpose the docking pose with CBR1 holo-structure to compare the binding modes of ASP9521 and the native ligand?
3, Could the authors test ASP9521 analogs with minor modifications in the enzymatic assays and summarize the SAR to confirm the docking pose?
4, Could the authors do an MD simulation on the CBR1 holo-structure and do a similar analysis, which could be a positive control for the stability validation?
Round 2
Reviewer 1 Report
The authors have satisfactorily answered to my previous comments. My recommendation is that the manuscript can be accepted in present form.